# Anti-Cancer Potential of Synthetic Oleanolic Acid Derivatives and Their Conjugates with NSAIDs

**DOI:** 10.3390/molecules26164957

**Published:** 2021-08-16

**Authors:** Wanda Baer-Dubowska, Maria Narożna, Violetta Krajka-Kuźniak

**Affiliations:** Department of Pharmaceutical Biochemistry, Poznan University of Medical Sciences, 4, Święcicki Street, 60-781 Poznan, Poland; maria.narozna@ump.edu.pl (M.N.); vkrajka@ump.edu.pl (V.K.-K.)

**Keywords:** oleanolic acid derivatives, Nrf2, NF-κB, NSAIDs, inflammation, oleanolic acid derivatives conjugates

## Abstract

Naturally occurring pentacyclic triterpenoid oleanolic acid (OA) serves as a good scaffold for additional modifications to achieve synthetic derivatives. Therefore, a large number of triterpenoids have been synthetically modified in order to increase their bioactivity and their protective or therapeutic effects. Moreover, attempts were performed to conjugate synthetic triterpenoids with non-steroidal anti-inflammatory drugs (NSAIDs) or other functional groups. Among hundreds of synthesized triterpenoids, still the most promising is 2-cyano-3,12-dioxooleana-1,9(11)-dien-28-oic acid (CDDO), which reached clinical trials level of investigations. The new group of synthetic triterpenoids are OA oximes. The most active among them is 3-hydroxyiminoolean-12-en-28-oic acid morpholide, which additionally improves the anti-cancer activity of standard NSAIDs. While targeting the Nrf2 and NF-κB signaling pathways is the main mechanism of synthetic OA derivatives′ anti-inflammatory and anti-cancer activity, most of these compounds exhibit multifunctional activity, and affect cross-talk within the cellular signaling network. This short review updates the earlier data and describes the new OA derivatives and their conjugates in the context of modification of signaling pathways involved in inflammation and cell survival and subsequently in cancer development.

## 1. Introduction

Oleanolic acid (3β-hydroxyolean-12-en-28-oic acid) is a pentacyclic triterpenoid, widely occurring in the plant kingdom, which includes edible and medicinal plants. The richest source of oleanolic acid (OA) are the leaves of the olive plant. Common culinary spices such as garden thyme and clove plants, as well as fruits, are also sources of OA [1,2]. OA and its natural derivatives possess several promising pharmacological activities, including hepatoprotective, anti-inflammatory, antioxidant, and, importantly, as a consequence of the above, anti-cancer activities. Since the elucidation of its biosynthesis and the commercialization of the first oleanolic acid-derived drug, the compound has become the subject of intensive studies [3].

Moreover, naturally occurring OA serves as a scaffold for additional modifications to achieve synthetic pentacyclic OA triterpenoids. Over the past years, a large number of triterpenoids have been synthetically modified in order to increase their bioactivity and their protective or therapeutic effects. Moreover, attempts to conjugate synthetic triterpenoids with non-steroidal anti-inflammatory drugs (NSAIDs) or other active components have been made.

Conjugation with NSAIDs is justified by the fact that hybridization is considered a leading synthetic trend in medicinal chemistry strategy on the one side and the inflammatory background of many types of cancer, such as HCC on the other. NSAIDs are supposed to play a role in cancer treatment and chemoprevention. However, most of these drugs act as COX-2 inhibitors, which, besides their beneficial effects of decreasing the risk of certain types of cancer, are well known for many unfavorable side effects. Fusing NSAIDs with the active oleanane structure, resulting in combined compounds, may enhance the biological effect with synergistic action and minimized toxicity or adverse reactions [4].

The key role in the inflammation process is played by transcription factor NF-κB, controlling the expression of genes such as COX-2 and *iNOS*, encoding cyclooxygenase-2 and inducible nitric oxide synthetase. Overexpression of COX-2, related to increased cell growth and invasiveness, is observed in human cancers. Downregulation of NF-κB negatively interferes with the Nrf2 signaling pathway, the major cellular defense against reactive oxygen species (ROS) and electrophilic species [5]. Modification of these signaling pathways is usually linked with the other pathways, leading to induction to apoptosis and inhibition of cell proliferation/survival. In recent years, excellent reviews have been published on synthetic triterpenoids, starting with the comprehensive work of the pioneers in this field, Liby and Sporn [6].

This short review updates the earlier data, and describes the new OA derivatives and their conjugates in the context of modification of signaling pathways involved in inflammation and cell survival and, subsequently, in cancer development.

## 2. Tumor Microenvironment and Signaling Pathways—Possible Targets of Oleanolic acid Derivatives

The critical role of the tumor microenvironment in the process of carcinogenesis is now widely accepted. Inflammatory cells, macrophages, neutrophils, or lymphocytes have the ability to generate highly reactive species, which directly damage DNA and can lead to the initiation of carcinogenesis and subsequently promote clonal growth of the initiated cells [7,8]. Moreover, inflammatory cells contribute to the process of angiogenesis [8]. Therefore, chronic inflammation, which promotes angiogenesis and is linked to the development of at least 30% of all cancers, is an important target of cancer prevention and therapy. Consequently, anti-inflammatory drugs are currently repurposing into oncological applications [7,9].

Two signaling pathways, NF-κB and Nrf2-ARE, activated by a variety of stimuli, play central roles in these processes. Moreover, although in different ways, NF-κB and Nrf2 transcription factors are involved in many steps of carcinogenesis through cooperation with multiple other signaling molecules and pathways [10]. As synthetic triterpenoids act, in most cases, as multifunctional agents, they may affect cross-talk between all these pathways.

The NF-κB protein family is composed of five structurally related members, including NF-κB1 (also named p50), NF-κB2 (also named p52), RelA (also named p65), RelB, and c-Rel, which mediate transcription of target genes such as COX-2 and *iNOS* by binding to a specific DNA element, κB enhancer, as various hetero- or homo-dimers [11]. Dimer p50/p65 is the most common active form of NF-κB. In this complex, the p65 subunit is responsible for initiating transcription, while the p50 subunit serves only as a helper in NF-κB DNA binding [11,12].

Normally, the NF-κB subunits are sequestered in the cytoplasm by inhibitory proteins, mainly IκB family members. Two signaling mechanisms are involved in NF-κB activation, canonical and non-canonical [11]. In the canonical pathway, activation of the IκB kinase (IKK) complex leads to phosphorylation and ubiquitination of IκB and subsequent degradation by the 26S proteasome. The released active dimers are translocated into the nucleus and activate target gene expression [13]. The non-canonical NF-κB activation does not involve IκBα degradation, but instead relies on the processing of the NF-κB2 precursor protein, p100, and is a selective response to a specific group of stimuli, including ligands of a subset of TNFR superfamily members, such as LTβR, BAFFR, CD40, and RANK [14].

After translocation into the nucleus, the NF-κB complex binds to a specific DNA target sequence (5′-GGGRNYYYCC-3′) [15], assembles with the basal transcriptional machinery, and might associate with other transcription factors, including AP-1 (c-Jun/c-Fos complex) and chromatin remodeling proteins, such as CREB-binding protein (CBP) and p300 [16]. A variety of factors can influence and activate NF-κB nuclear translocation, and thus will be responsible for the diverse and sometimes opposing roles of NF-κB as a pro-and an anti-inflammatory mediator [17]. In this regard, NF-κB is well known for providing cancer cells with a survival advantage by upregulating anti-apoptotic genes. Moreover, there is reciprocal cross-talk between NF-κB and autophagy in cancer. It can either promote or repress tumorigenesis, depending on the stimulus and the context [18].

The persistent activity of NF-κB in tumors is closely associated with the STAT3 transcription factor [19]. STAT3 enhances p300-mediated RelA acetylation, leading to the nuclear retention of NF-κB. Thus, persistently activated STAT3 maintains the constitutive NF-κB activity in tumor cells [20]. STAT3 signaling is a major intrinsic pathway for cancer inflammation due to its frequent activation in malignant cells, with a key role in regulating numerous genes crucial in inflammation-induced cancer and the tumor microenvironment. In this regard, aberrant STAT3 promotes uncontrolled cellular growth and survival through dysregulation of gene expression, including cyclin D1, c-Myc, Bcl-xL, Mcl-1, and thereby contributing to cells surviving. STAT3, besides the mechanism mentioned above, interacts with NF-κB at multiple levels; it is activated by several NF-κB-regulated gene products, e.g., interleukin-6 (IL-6). Thus, these two transcription factors regulate a multitude of genes essential for cancer-promoting inflammation. Similarly, although to a lesser extent, persistent activation of STAT5 in diverse human cancers increases proliferation, survival, angiogenesis, and metastasis [21].

Importantly, downregulation of NF-κB negatively interferes with the Nrf2 signaling pathway [5]. Nrf2 is responsible for cytoprotection, and the major cellular defense against ROS and electrophilic species [11]. However, it has also been demonstrated that Nrf2 is overexpressed in cancer cells, and may contribute to increased proliferation, invasion, and chemoresistance [22]. Nrf2 activators bound to a Kelch-like ECH-associated protein 1 (Keap1), causing a conformational change that renders Keap1 unable to promote Nrf2 degradation, allowing its translocation into the nucleus and binding to the ARE 5′-GTCACAGTGACTCAGCAGAATCTG-3′ (within the CBR1 promoter region) target sequence in DNA. Nrf2 can also be regulated by phosphorylation. Post-translational modification of Nrf2 by various protein kinases can affect the release of Nrf2 from the complex with Keap1, its nuclear translocation, and stability [23].

An interesting link between Nrf2 and NF-κB presents glycogen synthase kinase-3β (GSK-3β). GSK-3β can phosphorylate Nrf2 on tyrosine 568, resulting in activation of this factor. The same enzyme has been reported to be necessary for the full transcriptional activity of NF-κB, demonstrating that GSK-3β selectively supports the expression of a subset of genes activated by NF-κB-dependent proliferative signals [24,25].

The interconnection of signaling pathways such as Nrf2, NF-κB, and STAT3, and aberrations in their activation mechanisms, make it possible to use them as a target in cancer cells for potential chemotherapeutic and/or chemopreventive compounds. This approach fits in a new trend called anakoinosis [26], the name of which derives from the ancient Greek term for “communication.” The concept of anakoinosis is an alternative to conventional chemotherapy, which is usually based on a single target, or focused on a single area of the tumor. Unlike standard therapies, treatment protocols based on anakoinosis are less likely to lead to drug resistance.

Multifunctional synthetic triterpenoids seem to be good candidates for such alternative chemotherapeutics or chemopreventive agents.

## 3. Overview of Oleanolic Acid Derivatives—Chemical Approach

The synthesis of naturally occurring pentacyclic triterpenes with at least eight chiral centers, in particular OA, has been at the center of intense research. Chemical modifications of its structure to obtain synthetic OA derivatives are carried out to increase its potency, reduce toxicity, and increase bioavailability and solubility. Chemical modifications have led to a series of derivatives, such as 2-cyano-3,12-dioxooleana-1,9(11)-dien-28-oic acid (CDDO) which, in certain aspects, is over 200,000 times more potent than the parent OA [27,28]. Other promising structures are different esters and oximes [29,30].

OA is characterized by two methyl groups at C-4, one at C-8, one at C-10, one at C-14, and two at C-20. In addition, a carboxyl group is located at the C-17 position, and a double bond is present between C-12 and C-13. Moreover, the stereochemistry of the -OH group at the C-3 position has important physiological implications. In this regard, the less common 3α-OH isomers have different biological activities not shared by the more common 3β-OH isomers. These steric properties of the exocyclic methyl and other groups on natural triterpenoids are essential determinants of their activity and safety in pharmacological contexts and, therefore, became targets of modification [6].

The pioneering work of the Sporn and Liby group [6,31,32,33] and that which followed provided hundreds of synthetic OA derivatives, some of which were patented. Chemical modifications of OA at the C-3 hydroxyl group, at the C-12 to C-13 double bond, and at the C-28 carboxyl group led to new synthetic compounds that are many thousand-fold more active than the parent oleanolic acid, with useful pharmacokinetic and pharmacodynamic profiles. Modifications of the A and C rings to enones, of the C-2 substitution, of C-28 with a wide variety of esters and amides, and the modifications of the six-membered carbon ring A and E represent the most common modifications, leading to promising synthetic oleanolic acid derivatives (Figure 1) [30].

The most extensively explored OA derivatives are CDDO and its derivatives modified at the C-17 position: methyl ester (CDDO-Me; bardoxolone methyl), imidazolide (CDDO-Im), di-CDDO (nitrile at the C-17 position of CDDO; TP-225), and various amides (methyl amide, CDDO-MA; ethyl amide, CDDO-EA; and trifluoroethyl amide, CDDO-TFEA) [6,30]. The results of several studies have pointed out that the potential therapeutic activities of OA derivatives were correlated with their hydrophilicity. Although OA bears a hydroxyl group at the C-3 position and a carboxyl group at the C-17 position, the contribution of these two hydrophilic groups to the overall hydrophilicity of OA is limited. It might be assumed that hydrophilic moieties, coupled with certain long, flexible spacers, could improve OA′s hydrophilicity [34]. Therefore, a series of OA oximes were synthesized with different substituents at the C-17 position: basic carboxyl group, methyl ester, benzyl ester, or morpholide group, assuming this would provide higher hydrophilicity and bioavailability [35].

## 4. Docking Studies-Targeting Nrf2 and NF-κB

Taking into consideration the role of NF-κB and Nrf2 in inflammation and, subsequently, cancer development, attempts have been made to use docking to elucidate the interaction of some OA derivatives with these transcription factors. In this approach, the results are ranked by a scoring function, which usually has to be confirmed by redocking ligands that have been cocrystallized with the target [36].

Therefore, using several in silico techniques, Michael Sporn’s group recently demonstrated that CDDO-Im binds covalently to Keap1, the Nrf2 adaptor, by forming permanent Michael adducts with eight different cysteines, and acyl adducts with lysine and several tyrosine residues. Moreover, modeling studies suggested that the Tyr 85 adduct stabilizes the Keap1-Cul3 complex, thereby enhancing the potency of CDDO-Im. In contrast to CDDO-Me, CDDO-Im also showed the ability to covalently transacylate arginine and serine residues in glutathione S-transferase pi (GSTP) and cross-link them to adjacent cysteine residues [37]. On the other hand, the screening of seventy-seven natural and synthetic pentacyclic triterpenoids (PT) for Nrf2 stimulatory activity using the PASS (Prediction of Activity Spectrum of Substances) software, followed by in silico molecular docking against 16-mer Nrf2 peptide-binding site on Keap1, showed that Nrf2 stimulatory PTs dock on this binding site on Keap1, and may exert their biological activities by interfering with the Keap1 and Nrf2 binding. Interestingly the natural PTs were more promising than the most potent synthetic derivatives of oleanolic acid, such as CDDO, CDDO-Me, and CDDO-Im in this model [38].

Comparison of docking scores of OAO derivatives with diclofenac (DCL) to the small ligand-binding C-terminal Kelch domain of the human Keap1 (PDB entry: 4XMB) pointed the conjugate of 3-hydroxyiminoolean-12-en-28-oic acid morpholide and its protonated analog to be more potent and able to interact with the chemical environment within this cavity. The formation of several bonds between the DCL-OAO derivatives conjugates and amino acids within the cavity of the C-terminal Kelch domain of the Keap1 protein was demonstrated. More detailed analysis revealed an interaction with Arg415 and Gln530 within the tested cavity as the most significant for OAO derivatives with DCL, particularly the most active morpholide derivatives [39].

In the case of NF-κB, an additional difficulty in docking studies is the fact that these transcription factors form, as it was described above, a family of homo- and heterodimeric proteins resulting from the association of five different subunits.

Therefore, different subunits, or NF-κB inhibitory proteins, such as IκB or their relevant kinases, might be the docking targets. OA bearing an α-d-mannose moiety at C28 was the subject of docking studies targeting the NF-κB2 (p52) subunit after showing the significant cytotoxicity against colon cancer cells and inhibition of G1-S transition and inducing apoptosis. The results were consistent with its biological effect, and pointed out the binding of this derivative to NF-κB2, in contrast to parent OA, not showing such activity. Moreover, the detailed analysis of the interactions revealed that this derivative shares the hydrophobic interactions of the triterpenic aglycone with Lys119 and Gln157 and the salt bridge between the carboxyl group at C28 and an Arg residue [40].

In another screening study, 80 PTs were subjected to docking on the binding pocket of the IκB kinase subunit beta (IKKβ). The results showed that 56 of the tested ligands had docking scores of less than −1.50. The residues 85–101 of dimeric NF-κB essential modulator (NEMO) formed a flat slit, paving the way to two broad and extensive IKK-binding pockets, each pocket being occupied by the IKK peptide [41]. Interestingly, among the ligands showing a low docking score was CDDO-Im, although early studies using both cell-free and cellular assays showed that this compound is a direct inhibitor of IKKβ, and thereby inhibits binding of NF-κB to DNA and subsequent transcriptional activation [42].

The above examples indicate that, although the docking approach might be useful in the initial screening of oleanolic acid synthetic derivatives, their results might not be confirmed in biological systems. Moreover, the discrepancy between the systems might occur.

## 5. Biological Activity of Selected Synthetic OA Derivatives Which May Affect Cancer Development

As mentioned in previous sections, the most promising and, therefore, extensively studied was, and still is, CDDO and its modifications. These compounds were shown to interfere with many pathways involved in inflammation, oxidative stress, and cell proliferation.

Consequently, although they originally were developed as anti-inflammatory agents, they also became promising candidates for cancer chemotherapeutics and/or chemopreventive agents. In this regard, CDDO inhibited growth and induced cell cycle arrest and apoptosis in breast cancer cell lines [30]. In addition, CDDO induced apoptosis in leukemia cell lines [43] and induced differentiation of human myeloid leukemia cells [44]. As a potent anti-inflammatory agent, it suppressed the activity of the inflammatory mediators, inducible nitric oxide synthase (iNOS), and cyclooxygenase- 2 (COX-2) [45].

The CDDO C-28 modified derivative CDDO-Me (bardoxolone methyl), considered so far as the most active synthetic triterpenoid, induced apoptosis in human lung cancer cells [33]. In primary blast cells from leukemia patients, it inhibited the expression of anti-apoptotic, proliferative, and angiogenic gene expression [46]. Similar effects were observed in the case of the imidazole derivatives of CDDO. This derivative not only reduced the proliferation of different human cancer cell lines, but also decreased tumor burden in murine models of melanoma, leukemia, and several others, including lung, breast, and prostate cancer [47]. Mechanistic studies have shown that CDDO and its derivatives inhibit NF-κB by suppressing IκBα kinase, and are considered highly potent activators of Nrf2 and inducers of phase II enzyme [48,49].

The biological effects of CDDO and CDDO-Me were shown to be concentration-dependent [32]. At nanomolar concentrations, CDDO and CDDO-Me have been shown to protect cells and tissues from oxidative stress by increasing the transcriptional activity of Nrf2. At concentrations higher than 100 nM, CDDO and CDDO-Me are able to modulate the differentiation of a variety of cell types, both tumor cell lines and primary culture cells, while at micromolar concentrations, these compounds exert an anti-cancer effect in multiple manners. For example, they induce extrinsic or intrinsic apoptotic pathways or autophagic cell death, inhibit telomerase activity, disrupt mitochondrial functions through Lon protease inhibition, and block the deubiquitylating enzyme, the ubiquitin-specific-processing protease-7 [47].

Several other signaling pathways were reported to be affected by CDDO-Me, such as inhibition of the phosphoinositide 3-kinase/protein kinase B1/mammalian target of the rapamycin (PI3K/Akt/mTOR) and Janus-activated kinase (JAK)/STAT pathways [50]. Moreover, CDDO-Me and CDDO-Im were able to bind GSK3β, which led to alterations in adhesion dynamics of cell motility [51].

As a multifunctional and relatively noncytotoxic compound, CDDO-Me is considered to be applied in the prevention and treatment of not only cancer, but also other diseases with an inflammatory background. Therefore, the therapeutic effects of CDDO-Me have been tested in Phase III clinical trials for chronic kidney diseases and Phase I/II clinical trials for solid tumors and lymphomas [32,33,45,52]. Unfortunately, the first was terminated due to heart-related adverse effects [53]. Thus far, overall, more than 30 clinical trials have been conducted. Although a maximally tolerated dose (900 mg/day) was established, and complete response in a patient with mantle cell lymphoma and partial response in a patient with anaplastic thyroid carcinoma was achieved, ongoing trials focused rather on the use of improved protocols with CDDO-Me for treating chronic kidney diseases or pulmonary hypertension, rather than cancer [47].

Other synthetic derivatives of OA with modification at the C-17 or C-28 positions of the OA scaffold were shown to be potent inhibitors of induced nitric oxide production and expression of COX-2. This activity was related to reduced activation of NF-κB (IkBα degradation), STAT3 phosphorylation, and Nrf2 activation [30]. Additionally, a series of C-17-heteroaryl derivatives of OA that can exert antioxidant activity and be used as inflammation modulators were patented [54]. Recently, OA oxime derivatives were synthesized, and their ability to modulate signaling pathways involved in inflammation and cancer development was evaluated.

In this regard, the OA oxime (OAO) derivative methyl 3-octanoyloxyiminoolean-12-en-28-oate presented better anti-inflammatory activity than that of OA, which was demonstrated by its anti-oedemic effects in rats with carrageenan-induced skin inflammation [29]. Furthermore, evaluation of the effect of OA oxime derivatives with different groups at the C-17 position on NF-κB activation in human hepatoma cells (HepG2) indicated that the specificity of OAO structure substitution might be crucial for NF-κB modulation and subsequent anti-inflammatory effects. In this regard, comparison with parent compound OA and its oxime clearly indicated that the substitution within the carboxyl group at position C-17 increases their ability to modulate the NF-κB signaling pathway. Furthermore, substitution with morpholine, followed by substitution with a methyl group proved to be the most effective, whereas substitution with a benzyl group in OA or OAO did not alter their activity in NF-κB modulation, at least in human hepatoma cells [35].

Moreover, these compounds enhanced the activation and expression of Nrf2 in immortalized human hepatocytes (line THLE-2) and hepatoma cells (line HepG2). In the latter cell line, the association between cytotoxicity and Nrf2 activation was observed. In this regard, OAO with a morpholide or methyl ester at the C-17 position activated Nrf2 to the greatest extent, and subsequently led to cell cycle arrest at G2/M, leading to increased apoptosis and a number of resting HepG2 cells [55].

To sum up, CDDO and its methyl ester and imidazole derivative are still the leading OA modifications, and the only ones which have reached the clinical trials level. However, several new options were shown. The example of OAO demonstrates they are worth further study as potential therapeutic and/or chemopreventive agents, at least in certain types of cancers.

## 6. Conjugation of Synthetic OA Derivatives May Enhance Their Anti-Cancer Potential

Several approaches have been made to conjugate OA derivatives with different compounds in either the C-3 or the C-28 positions. In this regard, CDDO amino acid methyl ester derivatives functionalized at C-28 were patented. Their activity in suppressing NO production and activation of Nrf2 was comparable to that of CDDO-ethyl amide [56]. Rhodamine B--OA bioconjugate at the C-28 position of the OA molecule reduced the viability of several tumor cells. Moreover, impairment of melanoma mitochondrial function was observed, in contrast to the effect observed in the non-tumor keratinocytes cell line [57]. High selectivity of conjugates with uracil, particularly in hepatoma HepG2 cells, in the induction of apoptosis (triggering caspase-3/9 activity), was claimed [58].

Biological evaluation of OA oximes, conjugated with succinic acid OA oximes with the same substituents at the C-17 position, identified 3-succinyloxyiminoolean-12-en-28-oic acid morpholide (SMAM) as the most potent modulator of NF-κB and STAT3. SMAM significantly reduced the expression and activation of NF-κB, as well as the nuclear protein level of the p65 subunit. This compound also reduced the expression and activation of STAT3 and STAT5A/B. The combined effect of SMAM on these transcription factors resulted in reduced expression of COX-2, *MYC*, and anti-apoptotic *BCL-XL* genes [30].

Considering the inflammatory backgrounds of several cancers, non-steroidal anti-inflammatory drugs (NSAIDs) are supposed to be useful in cancer treatment. Numerous reports concerning the cancer-protective effects of NSAIDs have been published. Now, emerging evidence indicates that such drugs may have activity not only in chemoprevention, but also in treatment [9]. However, most of these drugs act as COX-2 inhibitors which, besides beneficial activity, decreasing the risk of certain types of cancer, are well known for many unfavorable side effects. In addition, their long-term use is often associated with many serious cardiovascular, gastrointestinal, renal, and other side effects [59].

Conjugation of OA derivatives with NSAIDs may enhance their pharmacological activity and avoid NSAIDs′ side effects [60]. Therefore, OAO derivatives with NSAIDs in the C-3 position were designed and evaluated (Figure 2).

The results showed that the conjugate of 3-hydroxyiminoolean-12-en-28-oic acid morpholide with aspirin (ASP) exerted strong anti-inflammatory activity in rats. The administration of this compound lowered the blood concentration of interleukin-6 (IL-6) and the IL-6 transcript level in blood lymphocytes [61]. Subsequent in vitro assays in HepG2 human hepatoma cells demonstrated that conjugation of OAO with ASP led to enhanced downregulation of NF-κB expression and activation. Among the hybrids with ASP, 3-(2-acetoxy)benzoyloxyiminoolean-12-en-28-oic acid morpholide and 3-(2-acetoxy)benzoyloxyiminoolean-12-en-28-oic acid methyl ester were the most efficient. As was mentioned in the previous section, these differ in having a morpholide or methyl ester group at the C-17 position of the OAO molecule. COX-2 transcript and protein levels were also diminished after treatment with these compounds [35]. Interestingly, 3-(2-acetoxy)benzoyloxyiminoolean-12-en-28-oic acid morpholide conjugated with ASP also affected Nrf2 activation and expression.

This effect in HepG2 cells was less pronounced in comparison with non-conjugated OAO. In contrast, conjugation enhanced Nrf2 activation in normal hepatocytes of the THLE-2 cells line. These results indicated that OAO derivatives conjugated with ASP have the potential for application in liver cancer chemoprevention. OAOs themselves, particularly OAO substituted with morpholide, may be considered therapeutic agents, which may support conventional treatment strategy [55]. In contrast to OAO conjugates with ASP, OAO hybrids with indomethacin and diclofenac reduced activation and expression, not only NF-κB, but also Nrf2 in hepatoma cells, while in normal cells, increased activation of Nrf2 was still observed (Figure 3 and Figure 4) [39,62].

As mentioned earlier in this review, NF-κB negatively interferes with the Nrf2 signaling pathway, and often agents that suppress NF-κB signaling activate Nrf2 [63]. Although the Nrf2-ARE pathway plays a cytoprotective role in normal cells, Nrf2 is frequently overexpressed in cancer cells, including HCC, and may contribute to increased proliferation, invasion, and chemo radio-resistance [64,65,66]. Therefore, the reduced activation of Nrf2 and increased ROS production resulting from treatment with OAO conjugates with indomethacin or diclofenac may protect cancer cells against chemoresistance inhibition of the Nrf2-ARE pathway and, at the same time, exert a chemopreventive effect in normal hepatocytes.

Moreover, it was found that CDDO-Me protected human colon epithelial cells against radiation-induced DNA damage through activation of Nrf2. Therefore, it may protect the gastrointestinal tract against acute irradiation [50]. Thus, OAO conjugates with NSAIDs may increase therapeutic efficacy in cancer cells while protecting normal cells in their microenvironment. The final effect depends on the type and structure of the particular class of NSAIDs.

## 7. Conclusions

Synthetic derivatives of naturally occurring triterpenoids increase their anti-inflammatory and anti-carcinogenic potential. Among the hundreds of compounds, CDDO-Me is still considered the most promising anti-inflammatory and anti-cancer agent. However, the results of clinical trials are not conclusive. Moreover, an increase in heart failure events as a result of treatment with CDDO-Me is an obstacle to its clinical use. Therefore, searching for other OA derivatives or their combination is still worthwhile.

Although the synthetic triterpenoids exhibit multifunctional activity, induction of the Nrf2-ARE pathway is dominant for most of them. Increased activation of this pathway might be a problem in cancer cells, which are often characterized by Nrf2 over-expression. Therefore, OAO conjugates, which inhibit both Nrf2 and NF-κB in cancer cells, and induce Nrf2 in normal cells, seem to be interesting candidates as dual-action agents, both chemopreventive and therapeutic. However, as these suggestions are based only on preliminary results obtained in specific experimental models, further comprehensive studies are required to confirm the potential of such OAO–NSAIDs hybrids.

## Figures and Tables

**Figure 1 molecules-26-04957-f001:**
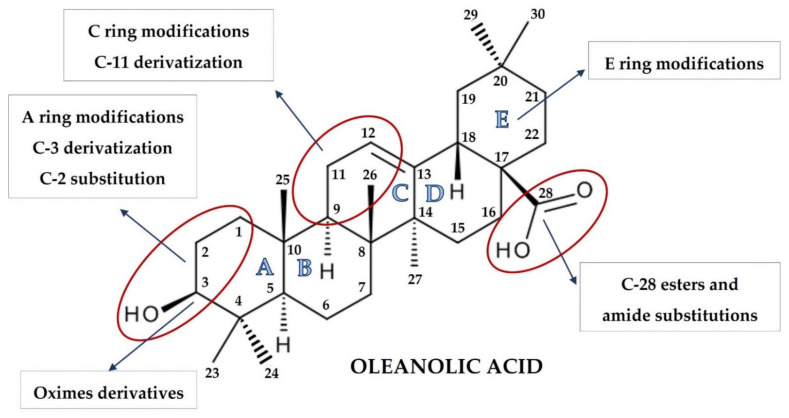
Sites of possible chemical modification of oleanolic acid structure that might lead to increased biological activity. Figure was created with BioRender.com by the authors.

**Figure 2 molecules-26-04957-f002:**
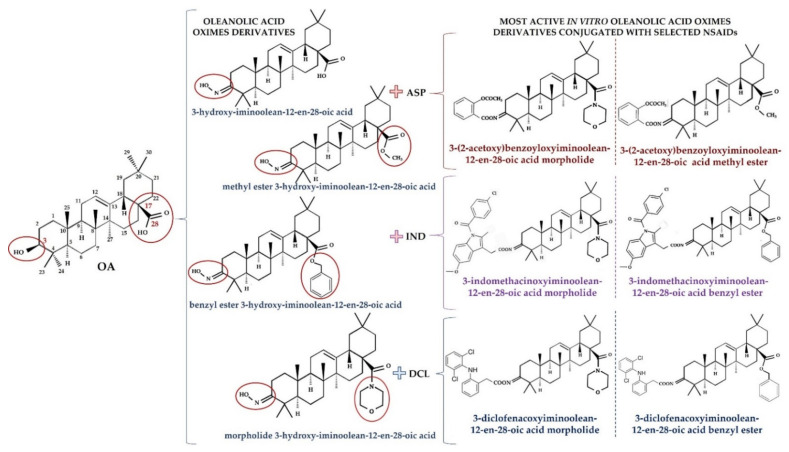
The structures of OAO derivatives and their conjugates with NSAIDs. Figure was created with BioRender.com by the authors.

**Figure 3 molecules-26-04957-f003:**
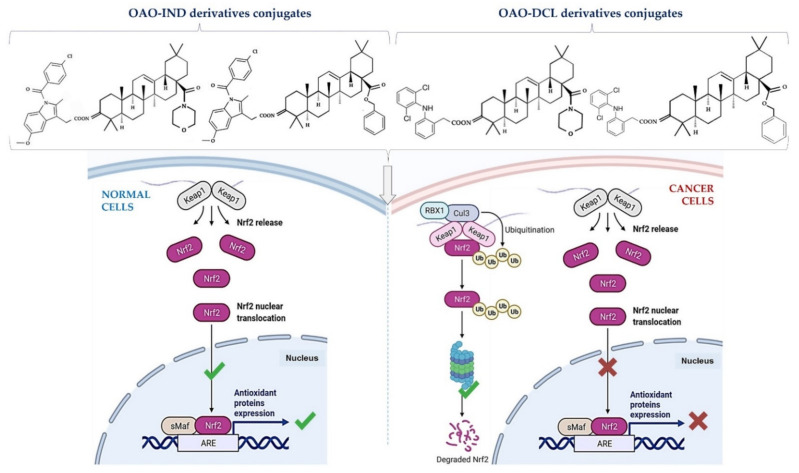
OAO-IND and OAO-DCL exert different effects on the Nrf2 pathway in normal hepatocytes and HCC cells. OAO–IND conjugates increase the activation of Nrf2 in normal hepatocytes by 25–55%, but inhibit its activation in HCC cells (by 30–40%) [62]. Similarly, OAO–DCL conjugates enhance the activation of Nrf2 in normal hepatocytes by 22–46% and inhibit activation of Nrf2 in HCC cells by 21–55% [39]. Figure was created with BioRender.com by the authors.

**Figure 4 molecules-26-04957-f004:**
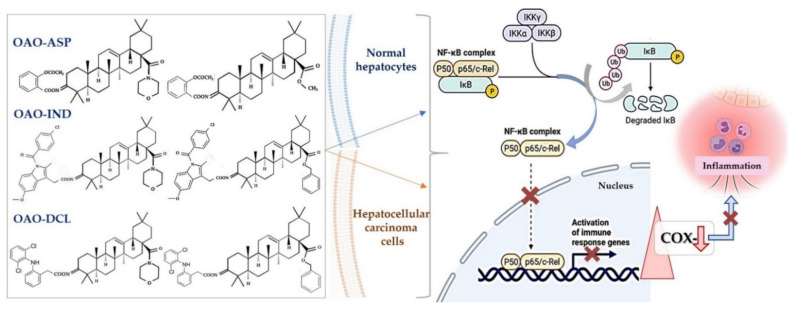
OAO–NSAIDs conjugates reduce NF-κB activation and expression of COX-2 in normal hepatocytes and HCC cells. OAO–NSAIDs inhibit activation of NF-κB in normal hepatocytes and in HCC cells by ~50–60% and ~40–50%, respectively [35,39,62]. Figure was created with BioRender.com by the authors.

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
