# Peer review of "Anti-Cancer Potential of Synthetic Oleanolic Acid Derivatives and Their Conjugates with NSAIDs"

_molecules, 2021, doi:10.3390/molecules26164957_

Round 1

Reviewer 1 Report

The authors present a review that includes anti-cancer potential of some synthetic oleanolic acid derivatives and their conjugates with NSAIDS.

In general, the manuscript is well written, and the discussion is interesting, however, presents one issue to be considered:

1.- It is well-known the importance of docking studies on ligands interactions and, in general, in drug design; however, information about this topic is missing in the review. The authors should consider including information about docking studies performed with synthetic oleanolic acid derivatives having ant-icancer or anti-inflammatory and their corresponding targets.

In conclusion, the review could be suitable for publication in Molecules if the authors attend the recommendations.

Author Response

Responses to Reviewers critical comments

Reviewer # 1

Q1. It is well-known the importance of docking studies on ligands interactions and, in general, in drug design; however, information about this topic is missing in the review. The authors should consider including information about docking studies performed with synthetic oleanolic acid derivatives having anti-cancer or anti-inflammatory and their corresponding targets.

A 1.  Following the Referee's suggestion, we have added the section entitled "Docking studies-targeting Nrf2 and NF-κB” in the revised manuscript (page 6-7).

Reviewer 2 Report

Minor revisión

  1. This article provides interesting experimental evidence of potential
    anti-cancer activity of new oleanolic acid derivatives conjugated with NSAIDS.
  2. The article is clear and easy to read. I would only ask the authors
    for better clarity in the Figures. I think it would be interesting for the authors to include numerical values of the biological activities shown by the derivatives of oleanolic acid.
  3. Line 171: change ( Fig. 1) to (Figure 1), no bold.
  4. For Figure 1, a darker tone of the structure is required and for the texts in the boxes.
  5. Line 294: change ( Fig. 2) to (Figure 2), no bold.
  6. For Figure 2, a darker tone of the structure is required and for all the indicated structures you must use a larger font size and for the compound names (it is difficult to read them)
  7. Line 318: (Fig. 3) to (Figure 3).
  8. For Figure 3, same comment as 4.
  9. Figure 4 is not mentioned in either of texts.
  10. For Figure 4, same comment as 4 and 6.
  11. Bibliographic references must be written in the journal format:

Journal Articles:

  1. Author 1, A.B.; Author 2, C.D. Title of the article. Abbreviated Journal Name Year, Volume, page range.

Author Response

Responses to Reviewers critical comments

Reviewer # 2

Q1. The article is clear and easy to read. I would only ask the authors for better clarity in the Figures. I think it would be interesting for the authors to include numerical values of the biological activities shown by the derivatives of oleanolic acid.

A1. Responding to the reviewer's comment, we improved the visual clarity and legibility of all the figures by increasing the font sizes, improving the color scheme and figures' sharpness. The numerical values of the biological activities of OAO derivatives are now presented in the the legends to Figures 3 and 4.

Q 2. Line 171: change ( Fig. 1) to (Figure 1), no bold.

A 2. The font in line 171 was changed into regular instead of bold and introduced the full name of Figure 1.  

Q 3. For Figure 1, a darker tone of the structure is required and for the texts in the boxes.

A 3. As requested, a darker tone of the structures and texts in the boxes were introduced.

Q 4. Line 294: change ( Fig. 2) to (Figure 2), no bold.

A 4.  As suggested by the reviewer, we have changed the font and introduced the full name of Figure 2.

Q 5. For Figure 2, a darker tone of the structure is required and for all the indicated structures you must use a larger font size and for the compound names (it is difficult to read them)

A 5. The suggested changes were introduced in Figure 2.

Q 6. Line 318: (Fig. 3) to (Figure 3).

A 6.  The full name of Figure 3 instead of the abbreviated form was introduced.

Q 7. For Figure 3, same comment as 4.

A 7.  As suggested by the reviewer, we have changed the fonts and introduced full name of Figure 3.

Q 8. Figure 4 is not mentioned in either of texts.

A 8. Figure 4 is  now mentioned in the text of the revised manuscript

Q  9. For Figure 4, same comment as 4 and 6.

A 9.  As suggested by the reviewer, we have changed the fonts and introduced the full name of Figure 4.

Q  10. Bibliographic references must be written in the journal format: Author 1, A.B.; Author 2, C.D. Title of the article. Abbreviated Journal Name Year, Volume, page range.

A 10. Bibliographic references were corrected according to the Molecules journal format.